# OpenDIVE: Streamlining Tractography Visualization

**Adam M. Saunders**[*1]  ADAM.M.SAUNDERS@VANDERBILT.EDU
[1] *Vanderbilt University, Nashville, TN, USA*
**Elyssa M. McMaster**[*1]  ELYSSA.M.MCMASTER@VANDERBILT.EDU
**Chris Rorden**[2]  RORDEN@MAILBOX.SC.EDU
[2] *University of South Carolina, Columbia, SC, USA*
**Johaan Kathilankal Jis**[3]  JKATHILA@STUDENTS.KENNESAW.EDU
[3] *Kennesaw State University, Georgia, USA*
**Minyi Sun**[1]  MINYI.SUN@VANDERBILT.EDU
**Adam Sadriddinov**[1]  MUKHAMMAD.SADRIDDINOV@VANDERBILT.EDU
**Lukas VanTilburg**[1]  LUKAS.T.VANTILBURG@VANDERBILT.EDU
**Michael E. Kim**[1]  MICHAEL.KIM@VANDERBILT.EDU
**Bennett A. Landman**[1,4]  BENNETT.LANDMAN@VANDERBILT.EDU
**Kurt G. Schilling**[4]  KURT.G.SCHILLING.1@VUMC.ORG
[4] *Vanderbilt University Medical Center, Nashville, TN, USA*

**Editors:** Under Review for MIDL 2025

## Abstract

Diffusion-weighted magnetic resonance imaging (DW-MRI) models provide a non-invasive method for mapping the structure of white matter. Despite the prolific availability of software tools to visualize DW-MRI, an MRI acquisition technique with increasing prevalence in deep learning applications, mapping meaningful summary statistics of white matter relationships can be a time-consuming process. We propose a Python-based command line tool to both standardize and improve accessibility to white matter models. We generate a standardized display of anatomical images to overlay diffusion models based on user input, and we provide a mechanism to display summary statistics (percent change, p-value, deep learning metrics, etc.) with a changeable color bar.

**Keywords:** Diffusion, tractography, visualization

## 1. Introduction

Diffusion-weighted magnetic resonance imaging (DW-MRI) models provide a non-invasive method for mapping the structure of white matter (Farquharson et al., 2013), (Westin et al., 2002). Tractography, the algorithmic process that reconstructs white matter pathways based on DW-MRI, can be performed on a whole brain or region-of-interest (ROI) basis, and yields metrics that help us characterize white matter integrity. Tractography ROIs, or bundles, are composed of individual streamlines – virtual representations of individual white matter pathways. Bundles are often used to summarize metrics across the region, such as average fractional anisotropy (FA) or mean diffusivity (MD) (Alexander et al., 2001), streamline characteristics such as average length, or properties such as surface area or volume (Yeh et al., 2018).

---

[*] Contributed equally

Table 1: The mapping of the percent change of bundle surface area on their corresponding brain regions is visible in Figure 1. Here, we visually understand the rates of change in these simulation bundle metrics.

| Region | Percent Change |
|---|---|
| CC 1 | -6.57% |
| CC 2 | -2.91% |
| OR Right | 8.50% |
| OR Left | 2.25% |
| AF Right | -2.30% |
| AF Left | 21.41% |
| CG Right | 7.07% |
| CG Left | -0.75% |
| MCP | 17.97% |

Tractography bundles can be used to further understand a variety of diseases and disorders, and have been the subject of many deep learning applications (Cai et al., 2023),(Wasserthal et al., 2018a), (Théberge et al., 2021). Many longitudinal studies include DW-MRI as part of their imaging protocol, which allows for the computation of both inter-subject and intra-subject deviations in white matter in contexts of diseases and disorders (Shock et al., 1984), (Jack et al., 2008), (Archer et al., 2023), notably Alzheimer's Disease, where the heterogeneous white matter changes impact different regions at different rates (Davatzikos et al., 2011). With the variety of tractography bundle summary statistics available and the heterogeneous nature of longitudinal change in the brain, there is demand for a visualization of these summary statistics in their corresponding bundle. However, widely used tractography visualization tools such as MI-Brain, Explore DTI, and MRtrix3 do not offer color mapping for tractography bundles in Python to summarize white matter bundle statistics. We introduce OpenDIVE: Open Diffusion Imaging Visualization for Everyone, version 1, as a streamlined Python package to leverage the Dipy toolbox (Garyfallidis et al., 2018) for tractography bundle visualization.

## 2. Demonstration

Brain development visualization provides an exciting opportunity to demonstrate a use case for the OpenDIVE tool. We select a subject from the MASiVar dataset (Cai et al., 2021b), available on OpenNeuro, who sat for longitudinal scans at age 5 and a year later at age 6. We preprocess images with PreQual (Cai et al., 2021a) and perform white matter segmentation and ROI-based tractography with TractSeg (Wasserthal et al., 2018a), (Wasserthal et al., 2018b), (Wasserthal et al., 2019), (Wasserthal et al., 2020). We assign bundle surface area for nine different bundles and interpret values with percent change over the year, overlaid on the b0 image, as show in Table 1. We use the `nifti2png` command with our desired MRI image and follow the instructions Wiki.

We show the mapping of the bundle surface area percent change on corresponding brain regions in an illustrative example in Figure 1. We can use the tool to visually understand the rates of change in these different bundle metrics.

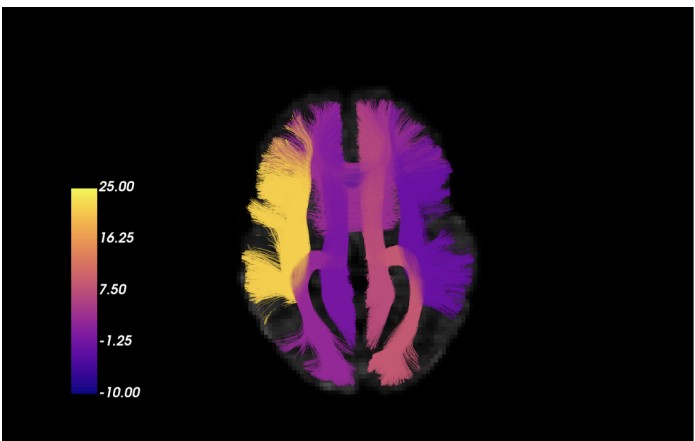

Figure 1: To emphasize significant changes bundle volume, we assign illustrative values to show capability and adjust the color bar to values between -10 and 25 percent and set the color bar option to True. The visualization gives us an intuitive understanding of the longitudinal changes in size of each white matter bundle.

## 3. Key Features and Future Directions

As shown in Figure 1, OpenDIVE can visualize tractography bundles with summary statistics. It can also display orientation distribution functions (ODFs) and diffusion tensors (DTI) in color-coded ROIs.

Future iterations of the OpenDIVE package will include further streamline visualization options, including shading and shaping (streamlines or tubes). As part of our project, we researched colorblind-friendly color map options and will include these in future versions. Those interested in contributing should visit our GitHub contributing page.

## Acknowledgments

We would like to thank the organizers of BrainHack Vandy 2025 for the opportunity to bring this idea to life. The Vanderbilt Institute for Clinical and Translational Research (VICTR) is funded by the National Center for Advancing Translational Sciences (NCATS) Clinical Translational Science Award (CTSA) Program, Award Number 5UL1TR002243-03. This work was conducted in part using the resources of the Advanced Computing Center for Research and Education at Vanderbilt University, Nashville, TN. We would like to acknowledge support from U24AG074855, 1R01EB017230, and P50HD103537.

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
