# OpenReview forum: "OpenDIVE: Streamlining Tractography Visualization"
_MIDL.io/2025/Short_Papers — MIDL 2025 - Short Papers_

### Official Review · Reviewer_9vyn · 2025-04-29

**Rating:** 4
**Confidence:** 4

**Summary:**

The authors introduce OpenDIVE, a Python tool for visualization of tractography bundles in DW-MRI with summary statistics.

**Strengths:**

The proposed tool OpenDIVE provides the function of color mapping for the statistical changes of tractography bundles, which is not supported in other widely-used visualization tools.

The tool also summarize and display longitudinal changes in white matter bundle statistics, which is helpful to understand various of disease and disorders.

**Weaknesses:**

The current function set of OpenDIVE is mainly designed for visualization for DW-MRI and its statistics, but there is not extra functions for user inputs and customization, e.g. measure the distance of certain white matter tract, or measure the detailed statistics in any user-selected ROI. Adding extra user interaction functions would be better for both visualization and research purposes.

---

### Decision · Program_Chairs · 2025-05-01

Accept